# Effects of Different Softening Processes on the Hardness and Quality of Thawed Neritic Squid (*Uroteuthis edulis*) Muscle

**Mark J. Grygier [1], Yu-Wen Fan [2] and Wen-Chieh Sung [1,2,*]** 

[1] Center of Excellence for the Oceans, National Taiwan Ocean University, Keelung 20224, Taiwan; mjgrygier@mail.ntou.edu.tw

[2] Department of Food Science, National Taiwan Ocean University, Keelung 20224, Taiwan; s5963618@gmail.com

\* Correspondence: sungwill@mail.ntou.edu.tw; Tel.: +886-2-2462-2192 (ext. 5129)

**Abstract:** While attempting to develop a soft, seafood-based product as a potential food item for senior citizens, we evaluated the effects of different softening methods on the hardness and quality of thawed muscular mantle tissue of the neritic squid (*Uroteuthis edulis*) versus controls. Comparisons were made among injection with proteolytic enzymes (papain, bromelain); soaking in these enzymes or in alkali solutions ($NaHCO_3$, $NaOH$); various soaking regimes combined with either orbital shaking under vacuum, ultrasonic processing, or ultrasonic cleaning; or hot-air drying and rehydration. Elderly panelists' sensory impressions of thawed and heat-sterilized squid mantle subjected to these treatments were recorded, together with the total volatile basic nitrogen (TVBN), pH, color, protein breakdown profile (SDS-PAGE), and histological characteristics of thawed squid mantle subjected to the same treatments but not heat-sterilized. TVBN measurements showed that squid mantle remained in a close-to-fresh state under all treatments except for hot-air drying. The pH and hardness decreased and the muscles turned white when soaked in either enzymes or alkali. Orbital shaking under vacuum while soaking in 0.3% papain for 12 h produced the softest product, the next softest being obtained by injection with 0.3% papain. After orbital shaking under vacuum in 1.0% papain, protein degradation was confirmed by SDS-PAGE, and broken muscle fibers were evident in histological sections. Sensory evaluation panelists with unimpeded chewing ability rated mushy, papain-treated squid mantle poorly. Soaking in 2% $NaHCO_3$ in an ultrasonic processor, following by washing out of the alkali, proved to be a better tenderizing method than either enzyme treatment or hot-air drying for neritic squid mantle intended for consumption by senior citizens.

**Keywords:** neritic squid; tenderization; bromelain; papain; food for seniors; palatability

## 1. Introduction

Neritic squid (*Uroteuthis edulis*, formerly called *Loligo edulis*; herein often simply called squid) is the major species in the squid fishery of Keelung, Taiwan, with an estimated take of 3780 metric tons worth NTD 530 million in 2013 [1]. Neritic squid contains high amounts of protein, taurine, omega-3 polyunsaturated fatty acids, iodine, selenium, vitamins $B_{12}$ and D, essential amino acids, and phospholipids [2]. Squid mantle contains a high amount of insoluble myostromin [3], making its meat tough and hard to chew compared to that of fish. Squid mantle muscle fibers, arranged both radially and circularly, are supported by connective tissue that is oriented in radial, circular, and longitudinal directions [4]. This has made it difficult to adapt squid mantle into a softer processed food for older people.

Squid mantle muscles can be tenderized by various processes to improve its texture. Sodium hydrogen carbonate (NaHCO₃) has been used as a tenderizer for both squid and meat in Chinese food for decades [5]. It loosens the meat's tissue structure by inducing the formation of a large number of small cavities. The proteolytic enzymes papain and bromelain are generally recognized as safe (United States Food & Drug Administration category "GRAS") for tenderizing collagenous and myofibrillar proteins [6]. Ultrasonic waves at appropriate intensity levels and frequencies can assist enzymes in inducing conformational changes in protein molecules but do not themselves affect the structural integrity of the tissue [7]. Ultrasonic treatment is an inexpensive, efficient, and low-polluting means of extracting and restructuring muscle protein. It has already been shown to shorten the period of alkaline soaking necessary for physical weakening of the mantle meat of jumbo squid (*Dosidicus gigas*), with a frequency of 25.6 KHz (low-frequency ultrasound) at a power of 186.9 W for 30.8 min being optimal [8].

In combination with exercise, increased protein intake could help to treat and limit the age-related physiological changes associated with declining muscle mass, strength, and function in elderly people [9]. Decreased hardness and increased hydration of meat protein brought about by soaking beef and pork in increasing concentrations of NaHCO₃ were positively correlated with elderly people's ease in consuming the meat [10]. Chicken breast soaked in 0.1–0.4 M NaHCO₃ also showed improved palatability and texture [5]. Furthermore, most seniors do not eat enough polyunsaturated fatty acids, especially omega-3 fatty acids, and this increases their risk of dementia, cardiovascular disease, and cognitive impairment [11]. Squid have the eighth highest docasahexaenoic acid (DHA) content among marine seafood commonly consumed in France [2].

Seafood is frequently subjected to hot-air drying [12], which causes structural and physico-chemical changes in the food and also leads to the degradation of proteins and concomitant changes in the microstructure of the food, which becomes more porous. As a result, the dried products are tough, shrunken, and brown, with low nutritive value. Upon rehydration, water is only partially, and not homogeneously, absorbed into it [13]. Despite these shortcomings, the drying–rehydration process can be used to soften food texture [14].

In the present study we evaluated the hardness and quality of neritic squid muscle treated by different softening methods, with a focus upon the development of tenderized food for seniors. Using thawed neritic squid as the raw material, we treated it with proteases and alkali solutions, with or without ultrasonic processing or vacuum orbital shaking, and also exposed it to hot-air drying and rehydration, always with comparison to untreated control samples. Changes in squid mantle toughness, tissue structure, pH, sensory evaluation of sterilized product by human subjects, and SDS-PAGE protein profiles were recorded and analyzed. The effects of such softening treatments have not been evaluated for neritic squid mantle muscle before, neither for freshly caught squid or, as here, frozen and thawed squid. Until now, comparisons of such parameters between fresh and thawed squid have been limited. Gokoglu et al. [15] reported that the initial freezing resulted in a significant less of hardness, and a sharp increase in cooking loss and total free amino acids, compared to fresh squid within 1 day, as measured both by instruments and sensory evaluation, but that there were no significant differences later on the hardness values of squid (*Loligo vulgaris*) that had been frozen up to 30 days.

## 2. Materials and Methods

### 2.1. Raw Materials and Controls

Neritic squid (*Uroteuthis edulis*) that had been frozen on board fishing boats after harvest were purchased from the Keelung Fishermen's Association (Keelung, Taiwan) on several occasions from June to September 2018. The squid were captured in the coastal region around Keelung during the same period, but the precise sites of capture are unknown. The frozen squid (>12 cm total length) were placed in running tap water until they had completely thawed, whereupon their heads, fins, viscera, and skin

of mantle were removed. The skinless mantles were cut into strips about 1.5 cm wide and soaked in a polyethylene bag in either 2% NaHCO$_3$ or 2% NaOH solution, or in deionized water as a control, for 4 days at 4 °C at a ratio of 1 g of squid per 1 mL of alkali solution. After soaking, the strips were washed with tap water until the surface of the muscle was no longer slippery [5]. Another set of similarly soaked mantle strips was loaded into an ultrasonic cleaner (DH200H, Yuantuo Technology, Taichung, Taiwan) and subjected to a frequency of 40 KHz at a power of 200 W for 30 min [8]. The sonicated strips were then immersed in boiling water for 1 min to inactivate all enzymes. Because squid mantle soaked in 2% NaOH had a strong, fishy, and alkaline off-odor, it could not be used in the sensory evaluation test described below; accordingly, most of the other trials described below only used the 2% NaHCO$_3$ samples.

### 2.2. Experimental Squid Samples

Squid mantle strips were soaked in either 0.5% or 1.0% papain, or 0.5% or 1.0% bromelain solution at a ratio of 1 g squid per 1 mL of enzyme solution and incubated in a 70 °C water bath for either 40 or 60 min, amounting to eight different enzyme treatments in all. Other mantle strips that had been soaked in a proportionally four times greater volume of the same enzyme solutions were loaded into an ultrasonic processor (UP500, ChromTech, New Taipei City, Taiwan) and subjected to a frequency of 20 KHz at a power of 150 W for 30 min [8]. The sonicated strips were then immersed in boiling water for 1 min to inactivate all enzymes. Other thawed squid strips were injected with the two aforementioned enzymes at a dosage of 1 mL of 0.1%, 0.2%, or 0.3% papain or bromelain solution per 10 g of mantle (six experimental treatments altogether), the solution being delivered perpendicular to the plane of the strip at sites approximately 1 cm apart. Injected and control samples were kept for 24 h at 4 °C [16]. Orbital shaking in vacuum was carried out using a vacuum desiccator (P14-1000240, Hondwen Co. Ltd, Taipei, Taiwan) with a polycarbonate top and polypropylene bottom enclosing a cylindrical chamber 240 mm wide and 311 mm deep, placed atop an orbital shaker (MS-NOR 30, Major Science, Saratoga, CA, USA). Shaken and unshaken (control) samples were kept for 4, 8, 12, and 16 h at 4 °C and then immersed in boiling water for 1 min [17].

For mixed treatments using both enzymes and mechanical stimuli, squid mantle strips were soaked in either 0.5% and 1.0% papain or bromelain solutions in a beaker at a ratio of 1 g of squid per 1 mL of enzyme solution and shaken as above for 4, 8, 12, or 16 h, then immersed in boiling water for 1 min to inactivate the enzyme [17]. Other samples of squid mantle were similarly soaked in enzyme solutions and then loaded into the aforementioned UP500 ultrasonic processor and subjected to a frequency of 40 KHz at a power of 200 W for 30 min. Again, enzymes were deactivated by immersion of the strips in boiling water for 1 min.

Hot-air drying was performed on a flat tray in a hot circulation exact oven (RHDM-452, Mandarin Scientific Co. Ltd, New Taipei City, Taiwan) at 50 °C or 60 °C in two different series of trials. Squid mantle strips were dried until a final moisture content of 10% had been reached. The hot-air dried squid was immersed in distilled water at 40 °C for up to 1 h, while removing portions at intervals of 10 min, and the rehydration index for each subsample was calculated as the quotient of the weight of the rehydrated material and its original 10% moisture-content weight [13].

The proximate composition of both raw and sterilized thawed neritic squid was determined according to the methods prescribed by the Association of Official Analytical Chemists [18].

### 2.3. Chromaticity Testing

Color of treated and control samples was measured at three different positions on squid mantle strips using a colorometer (TC-1800 MK-II, Tokyo Denshoku, Tokyo, Japan) calibrated with a white standard tile. Readings were expressed according to the "Hunter Lab" scale, where "L" measures lightness (100 being very white and 0 being black), "a" measures redness (+) to greenness (−), and "b" measures blueness (−) to yellowness (+) [19].

### 2.4. pH and Total Volatile Basic Nitrogen (TVBN)

Squid mantle strips without skin were homogenized in five times their volume of deionized water in a disperser (T18 basic, IKA, Staufen, Germany). The pH of the homogenate was measured with a pH meter (pH 510, Thermo Eutech, Singapore) [19]. Microdiffusion analysis of volatile nitrogen [20] was performed using 1 mL of trichloroacetic acid extract of squid mantle in a Conway microdiffusion dish with 1 mL saturated $K_2CO_3$ as the releasing agent and 1 mL 10% $H_3BO_3$ as the trapping agent with an indicator dye (methyl red). After diffusion for 90 min at 37 °C, the trapping agent was titrated to the original methyl red color with 0.1 N HCl. The volatile nitrogen was calculated following the method of Cobb et al. [20]. All analyses were conducted in triplicate.

### 2.5. Light Microscopy of Squid Mantle Muscle Tissue

A number of 2–3 mm thick strips of squid mantle subjected to different softening techniques as outlined above were fixed with 10% formalin at 4 °C for 24 h. Dehydration and embedding were performed according to [8]. Samples were dehydrated in 70%, 90%, and 100% ethyl alcohol for 1.5 h each to remove free water in the fixed tissue, then subjected to two changes of xylene for 1 h each, infiltration in melted paraffin at 60 °C for 30 min, and then embedding in paraffin for 24 h. The block face was trimmed to $5 \times 5 \times 3$ mm. Thick sections (6 μm) were prepared with a rotary microtome (Tissue-Tek VIP5 JR, Sakura, Tokyo, Japan) using a steel knife. Tissue sections mounted on glass slides were dried at 40 °C [21] and immersed in two changes of xylene for 10 min each to dissolve the paraffin, then immersed in ethyl alcohol of decreasing concentration: 100%, 95%, 90%, 85%, 80%, and 75% for 5 min each. After washing with distilled and double distilled water, the slides were immersed in a saturated hematoxylin solution for 20 s and washed under tap water for 15 s. The slides were then stained by immersion in eosin (Thermo Fisher Scientific, Kalamazoo, MI, USA) for 45 s the excess being washed out in tap water. The slides were then dehydrated in ethyl alcohol of increasing concentration −75%, 80%, 85%, 90%, 95%, and 100%, for 10 s each, and then in xylene for 10 sec [21]. Then, glass cover slips were applied with mounting medium (Micromount, Leica Biosystems Richmond, Inc., Richmond, IL, USA). Observations were done with a compound photomicroscope (BX 53 Upright Microscope, Olympus, Melville, NY, USA) at 200× and 400× equipped with a digital camera (BX 53 Upright Microscope, Olympus, Melville, New York, USA).

### 2.6. Sodium Dodecyl Sulfate-Polyacrylamide Gel Electrophoresis (SDS-PAGE)

Squid mantle protein was extracted according to [22]. Chopped squid mantle muscle (1.5 g) was homogenized in 30 mL of 0.02 M Tris-HCl at pH 8.0 with 0.2 g β-mercaptoethanol, 0.2 g sodium dodecylsulfate, and 0.024 g urea (Merck, Hoenebrunn, Germany) and centrifuged at 15,000× *g* for 30 s (CR21G, Hitachi Koki, Tokyo, Japan) followed by continuous shaking for 3 h at 100 rpm in an ice bath. Insoluble material in the homogenates was removed by centrifugation at 16,670× *g* for 45 min at 4 °C. The supernatant was analyzed for myofibrillar protein degradation by SDS-PAGE. An aliquot of 20 μL from each treated sample was subjected to SDS-PAGE under reducing conditions. After electrophoresis (Gene Power Supply SPS 200/400 apparatus, Pantech, Taipei, Taiwan), the proteins in the gels were stained with Coomassie Brilliant Blue R-250 [22].

### 2.7. Texture Analysis

Hardness of experimental and control squid mantle samples was measured using a texture analyzer (TA-XT2, Stable Micro Systems, Godalming, United Kingdom) equipped with a 10 mm diameter cylinder probe (P/0.5) (pre-test speed: 2.0 mm/sec; test speed: 2.0 mm/sec; post-test speed: 10.0 mm/sec; strain: 50%; trigger force: 10 g). The compressive force rises with sample hardness. Tests were performed with three replicates of each sample centered under the probe [16].



*2.8. Sensory Evaluation*

Sensory evaluation of pouch samples of 250 g of sterilized squid mantle was carried out to obtain subjective information on color, texture, appearance, flavor, and fishy smell from untrained panelists on the basis of ranking tests and an index of overall acceptability. Sterilization was accomplished at 121 °C for 20 min in an autoclave (TM-322, Tomin Medical Equipment Co., Ltd., New Taipei City, Taiwan). Before serving, the pouches were reheated by immersion in boiling water for 5 min. The 51 panelists consisted of 12 male and 39 female students of the General Education Center of National Taiwan Ocean University ranging in age from 55 to 82 (mean age 67), 21 of them being over 70. Each panelist received a total of four test samples representing the three treatments described above (papain injection, papain soaking in a vacuum orbital shaker, $NaHCO_3$ soaking with ultrasonic cleaning) plus an untreated control sample. The samples, each consisting of two squid strips accompanied by two dips of soy sauce, were coded with three random digits and presented to the panelists in random order. Panelists were instructed to evaluate each of the aforementioned attributes by ranking it from "1 = extremely like" to "4 = extremely dislike". Each data point represents the sum of the rankings by all 51 panelists.

*2.9. Statistical Analysis*

The data were analyzed with the SPSS statistics package for Windows, version 12 (SPSS Inc., Chicago, IL, USA). Ranking tests for the aforementioned sensory attributes, plus overall acceptability, were carried out using Friedman's analysis of variance (ANOVA) and Duncan's multiple range test in order to detect differences between treatments at a 5% significance level ($p < 0.05$).

## 3. Results and Discussion

*3.1. Effects of Alkali Soaking and Ultrasonic Cleaning on Hardness and pH of Squid Muscle*

Soaking in an alkali solution resulted in softening of test samples of squid mantle, with the effect being greater in 2% NaOH than in 2% $NaHCO_3$ ($p < 0.05$) (Table 1). The pH of squid mantle soaked in 2% NaOH (9.93) was significantly higher than that of samples soaked in 2% $NaHCO_3$ (9.16) or a control sample (7.59). As was noted above, owing to the foul odor of squid mantle soaked in 2% NaOH, subsequent trials only used squid mantle soaked in 2% $NaHCO_3$.

A significant softening effect was also found when alkali soaking was combined with ultrasonic cleaning, and the soaking time could be reduced from 4 days to 30 min. Ultrasonic cleaning thus proved to be an inexpensive, time-saving adjunct to the process of softening squid muscle in alkali solution. In one previous study [8], ultrasonic treatment alone was demonstrated to tenderize squid muscle, a significant decrease in hardness being associated with damage to mantle muscle fibers. Ultrasonic treatment degraded large-Dalton muscle protein components including collagen, but did not alter the pH of squid muscle. In our study, however, squid mantle was not tenderized by ultrasonic treatment alone, whether performed by an ultrasonic cleaner or an ultrasonic processer (Table 2).

The iso-electric point (PI) of chicken breast meat is in the pH range of 5.0–5.4, and its water-holding capacity is lowest in the same pH range [5]. The volume available within the myofibrils for holding water increased with higher pH because most of the side-chain groups are negatively charged and are thus expelled from the fibers. The myofibrils' capacity for holding water also increased with exposure to $NaHCO_3$ due to a charge imbalance. We assume that many of the same considerations apply to squid mantle meat. Any increase in pH away from the iso-electric point will increase the water-holding capacity of the myofibrils and make the squid muscle not only more tender, but also juicier.

*3.2. Enzyme Treatment and Color*

Exposure to proteolytic enzymes had the effect of decreasing the hardness of squid muscle as the concentration of either tested enzyme increased, and papain was a more effective tenderizer than bromelain (Tables 2 and 3). Injection with 0.3% papain resulted in a significant decrease in hardness

compared to the controls (Table 3). Enzyme soaking produced a smaller effect than enzyme injection because with soaking alone, only the surface of the test strips of squid mantle was affected. To make up for this, the soaking concentration was increased to 0.5% and 1.0% in subsequent trials. The lowest hardness (4.68 N/m$^2$) was obtained when squid mantle was injected with 1.0% papain and agitated in a vacuum orbital shaker (Table 4), and enzyme-soaked squid mantle treated with an ultrasonic processor was softer than that subjected to ultrasonic cleaning (Table 4). The cavitation effect of the ultrasonic processor used at 20 KHz on squid muscle soaked in 1.0% papain was favorable for enzyme conformational changes, resulting in an enhancement of enzyme activity (Table 4). At higher ultrasound frequencies, such as 40 KHz at 200 W, papain activity might have been inhibited by the violent collapse of too many cavitation bubbles in the enzyme solution and the resulting generation of excessive heat (Table 4) [7].

In an earlier study of meat from jumbo squid, the higher the air-drying temperature, the lower the rehydration index became [14], probably due to heat-related changes in the protein matrix. In the present study, there was no significant change in the hardness of dried–rehydrated squid mantle prepared at 50 °C and 60 °C compared with raw, untreated squid mantle (Table 5), but the rehydration rate was indeed higher for squid mantle dried at 50 °C (2.51) than at 60 °C (1.94). Higher drying temperatures denature proteins and myofibrils, resulting in shrinkage in muscle cells and the observed lower rehydration index for squid fillets [13]. After realizing this, we only used squid mantle dried at 50 °C in subsequent trials.

Compared with controls, an obvious increase in whiteness and decrease in redness and yellowness were found in squid mantle treated by soaking in alkali or proteolytic enzyme solutions (Table 6). In samples subjected to hot-air drying and rehydration, "L" decreased significantly whereas "a" and "b" increased significantly ($p < 0.05$) compared with the controls, as the mantle strips became darker, redder, and yellower as drying proceeded (Table 6). This color change may have been due to the Maillard reaction, oxidation or concentration of pigment, or denaturation of myoglobin [23], but in any case is an indication of sample browning [24].

Raw thawed squid mantle had a moisture content of 80.9%, together with 14.4% protein, 1.1% fat, 3.4% ash, and 0.2% carbohydrate, and these proportions did not change appreciably after sterilization at 121 °C for 20 min ($p > 0.05$) (Table 7). In contrast, the moisture content of squid strips soaked in 1.0% papain in a vacuum orbital shaker, which, before sterilization, already contained less moisture and protein than raw squid, decreased further to 78.2% after sterilization while the protein content increased to 16.5% (Table 7). It may have been that collagen was degraded more severely, that is, it was more hydrolyzed and denatured when exposed to papain, thus reducing its water-storage capacity [3].

**Table 1.** Hardness of neritic squid muscle soaked in alkali and subjected to ultrasonic cleaning.

| | Hardness ($\times 10^4$ N/m$^2$) | | |
|---|---|---|---|
| | Control | 2% NaHCO$_3$ | 2% NaOH |
| Soaked at 4 °C for 4 days | 18.62 ± 1.18 [a] | 5.12 ± 0.63 [b,B] | 0.26 ± 0.01 [c,B] |
| Soaked for 30 min with ultrasonic cleaning | 18.91 ± 0.39 [a] | 4.94 ± 0.37 [b,B] | 1.60 ± 0.33 [c,B] |
| Soaked for 30 min without ultrasonic cleaning | | 10.10 ± 0.88 [a,A] | 6.18 ± 0.46 [b,A] |

Expressed as mean ± standard deviation (*n* = 3). Means in the same row or column followed by different small letter [(a–c)] and capital letter [(A,B)] superscripts, respectively, are significantly different (*p* < 0.05).

**Table 2.** Hardness of neritic squid muscle soaked in bromelain (B) and papain (P) enzymes and subjected to various treatments, versus untreated controls.

| | Hardness ($\times 10^4$ N/m$^2$) | | | | |
|---|---|---|---|---|---|
| | Control | B 0.5% | B 1.0% | P 0.5% | P 1.0% |
| **Soaking time** | | | | | |
| 40 min | 18.77 ± 1.88 [a,A] | 14.61 ± 1.98 [b,B] | 14.20 ± 1.69 [b,B] | 12.94 ± 1.44 [b,c,B] | 10.11 ± 0.94 [c,B] |
| 60 min | 18.99 ± 1.16 [a,A] | 16.35 ± 1.13 [a,b,A] | 16.20 ± 1.56 [a,b,A] | 14.81 ± 1.57 [b,A] | 14.23 ± 1.44 [b,A] |
| **Vacuum Orbital Shaking** | | | | | |
| 4 h | 12.80 ± 0.42 [a,A] | 9.16 ± 1.34 [b,B] | 8.88 ± 0.71 [b,B] | 8.68 ± 0.70 [b,B] | 8.03 ± 0.97 [b,B] |
| 8 h | 12.70 ± 0.15 [a,A] | 8.29 ± 1.93 [b,C] | 8.56 ± 0.67 [b,B] | 8.56 ± 0.31 [b,B] | 7.69 ± 0.75 [b,C] |
| 12 h | 12.61 ± 0.08 [a,A] | 9.25 ± 0.72 [b,B] | 7.14 ± 1.57 [c,C] | 5.14 ± 1.45 [d,C] | 4.68 ± 0.24 [d,E] |
| 16 h | 12.31 ± 0.30 [a,A] | 9.30 ± 1.09 [b,B] | 8.84 ± 0.64 [b,B] | 8.65 ± 0.72 [b,B] | 6.11 ± 3.20 [b,D] |
| 16 h enzyme control | | 10.20 ± 0.82 [a,A] | 10.12 ± 0.34 [a,A] | 10.07 ± 0.19 [a,A] | 10.01 ± 0.12 [a,A] |
| **Ultrasonic Processor** | | | | | |
| 30 min | 15.04 ± 0.97 [a] | 9.76 ± 1.07 [b,B] | 8.15 ± 1.45 [b,c,B] | 8.26 ± 1.32 [b,c,B] | 7.23 ± 0.31 [c,B] |
| enzyme control | | 13.53 ± 0.63 [a,A] | 13.01 ± 0.81 [a,A] | 13.15 ± 1.00 [a,A] | 12.62 ± 0.35 [a,A] |
| **Ultrasonic Cleaning** | | | | | |
| 30 min | 18.91 ± 0.39 [a] | 11.77 ± 1.55 [b,B] | 11.15 ± 0.68 [b,B] | 11.81 ± 1.20 [b,B] | 11.06 ± 0.20 [b,B] |
| enzyme control | | 13.53 ± 0.63 [a,A] | 13.01 ± 0.81 [a,A] | 13.15 ± 1.00 [a,A] | 12.62 ± 0.35 [a,A] |

Expressed as mean ± standard deviation (*n* = 3). Means in the same row or column followed by different small letter [(a–c)] and capital letter [(A–E)] superscripts, respectively, are significantly different (*p* < 0.05). Enzyme control: soaked in enzyme solution but not otherwise treated.

**Table 3.** Hardness of neritic squid muscle either injected with or soaked in bromelain (B) and papain (P) enzymes, versus untreated controls.

| | Hardness ($\times 10^4$ N/m$^2$) | | | | | | |
|---|---|---|---|---|---|---|---|
| | Control | B 0.1% | B 0.2% | B 0.3% | P 0.1% | P 0.2% | P 0.3% |
| Injection | 17.75 ± 1.84 [a,A] | 8.65 ± 0.94 [b,B] | 6.85 ± 1.08 [c,B] | 5.93 ± 0.31 [c,d,B] | 6.49 ± 0.77 [c,d,B] | 5.42 ± 0.31 [d,e,B] | 4.74 ± 0.20 [e,B] |
| Soaking | 18.75 ± 1.84 [a,A] | 17.52 ± 1.05 [b,A] | 16.77 ± 0.10 [b,c,A] | 15.85 ± 0.16 [d,e,A] | 17.10 ± 0.10 [b,c,A] | 16.31 ± 0.25 [c,d,A] | 15.23 ± 0.17 [e,A] |

Expressed as mean ± standard deviation (*n* = 3). Means in the same row or column followed by different small letter [(a–e)] and capital letter [(A,B)] superscripts, respectively, are significantly different (*p* < 0.05).

**Table 4.** Hardness of neritic squid muscle treated with different softening methods, versus untreated controls.

| | Control | Hot-Air Drying | Alkali Soaking | Vacuum Orbital Shaker | Enzyme | | Ultrasonic Cleaning | | Ultrasonic Processor |
| --- | --- | --- | --- | --- | --- | --- | --- | --- | --- |
| | | | | | Injection | Soaking | Enzyme | Alkali Soaking | With Enzyme |
| Hardness ($10^4$ N/m²) | 19.58 ± 0.41[a] | 12.06 ± 3.48 [b] | 5.12 ± 0.63 [d] | 4.68 ± 0.24 [d] | 4.74 ± 0.20 [d] | 10.11 ± 0.94 [c] | 11.06 ± 0.20 [c] | 4.94 ± 0.37 [d] | 7.23 ± 0.31 [c,d] |

Expressed as mean ± standard deviation ($n = 3$). Values followed by different superscript letters within each row are significantly different ($p < 0.05$).

**Table 5.** Hardness of neritic squid muscle rehydrated at two temperatures after hot-air drying.

| | Hardness ($\times 10^4$ N/m²) | | |
| --- | --- | --- | --- |
| | Control | 50 °C | 60 °C |
| Raw | 34.71 ± 0.86 [a,A] | 34.42 ± 1.56 [a,A] | 34.68 ± 1.0 [a,A] |
| After sterilization at 121 °C for 20 min | 18.91 ± 0.39 [a,B] | 12.06 ± 3.48 [b,B] | 12.76 ± 2.54 [b,B] |

Expressed as mean ± standard deviation ($n = 3$). Means in the same row or column followed by different small letter [(a–c)] and capital letter [(A,B)] superscripts, respectively, are significantly different ($p < 0.05$). Raw: unblanched; control: blanched.

**Table 6.** Color of neritic squid muscle treated by different softening methods, versus raw and untreated controls.

| Color Values | Raw Squid | Control Sterilized Sample | S | SU | EI | VOS | UP | RAHD |
| --- | --- | --- | --- | --- | --- | --- | --- | --- |
| L | 71.31 ± 0.61 [a] | 84.12 ± 0.11 [c] | 93.16 ± 0.22 [e] | 88.75 ± 0.84 [d] | 94.49 ± 0.29 [e] | 95.73 ± 0.01 [f] | 93.86 ± 0.20 [e] | 80.89 ± 0.25 [b] |
| a | −15.18 ± 0.10 [a] | −8.44 ± 0.06 [d] | −13.59 ± 0.33 [b] | −9.18 ± 0.35 [d] | −11.37 ± 0.02 [c] | −11.05 ± 0.03 [c] | −13.20 ± 0.12 [b] | −5.03 ± 0.93 [e] |
| b | 20.03 ± 0.18 [a] | 38.19 ± 0.08 [c] | 35.81 ± 1.97 [c] | 32.78 ± 0.17 [b] | 32.72 ± 1.36 [b] | 25.45 ± 0.11 [b] | 34.39 ± 0.73 [c] | 46.05 ± 0.27 [d] |

Expressed as mean ± standard deviation ($n = 3$). Values followed by different letters within each row are significantly different ($p < 0.05$). S: soaked in 2% NaHCO₃; SU: soaked in 2% NaHCO₃ with ultrasonic cleaning; EI: injected with 0.3% papain solution; VOS: soaked in 1.0% papain solution in vacuum orbital shaker; UP: soaked in 1.0% papain solution in ultrasonic processor; RAHD: rehydrated after hot-air drying.

**Table 7.** Proximate composition of neritic squid.

| | Moisture (%) | Crude Protein (%) | Crude Fat (%) | Ash (%) | Carbohydrate * (%) |
| --- | --- | --- | --- | --- | --- |
| Raw squid | 80.90 ± 0.07 [a] | 14.44 ± 0.19 [a] | 1.10 ± 0.39 [a] | 3.39 ± 0.49 [a] | 0.17 ± 0.22 [a] |
| Control | 80.35 ± 0.61 [a] | 14.46 ± 0.06 [a] | 1.28 ± 0.17 [a] | 3.79 ± 0.03 [a] | 0.12 ± 0.15 [a] |
| VOS | 78.19 ± 0.05 [b] | 16.50 ± 1.33 [b] | 1.36 ± 0.10 [a] | 3.85 ± 1.05 [a] | 0.10 ± 0.54 [a] |

Expressed as mean ± standard deviation ($n = 3$). Values followed by the different superscript letters within each column are significantly different ($p < 0.05$). * Carbohydrate: estimated as the remainder after subtracting the other components. Control: untreated sterilized sample; VOS: soaked in 1.0% papain solution in vacuum orbital shaker.

### 3.3. pH and Total Volatile Nitrogen (TVBN)

Enzyme treatment and hot-air drying both significantly lowered the pH of squid mantle (Figure 1). In other studies also, a significantly lower pH has been recorded for squid samples treated with papain or bromelain, compared to control samples [14]. This may be due to a release of free amino acids following exposure of squid muscle to these proteolytic enzymes. Lowered pH values have also been reported in enzyme-treated pork, giant catfish, and chicken [25].

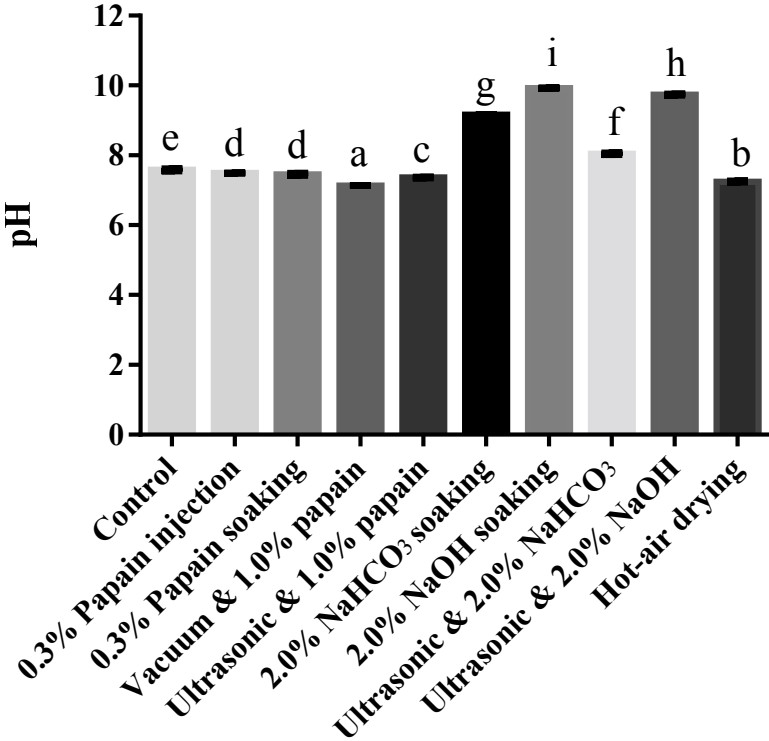

**Figure 1.** pH of neritic squid muscle treated with different softening methods, versus untreated controls. Different small letters (a–i) indicate significant differences ($n = 3$; $p < 0.05$).

TVBN has been used as an indicator of fish quality or freshness [14]. Its initial value in the present study's frozen squid mantle was 5.10 mg/100 g sample, a value comparable to those reported for raw fish previously [14]. Freshly caught, unfrozen squid were not available for analysis, but because only pre-frozen squid are sold on the open market, our use of raw, thawed squid as controls was justified. For squid mantle soaked in 2% $NaHCO_3$ solution for 4 days, or injected with 0.3% papain solution and under refrigeration at 4 °C for 24 h, or soaked in 1.0% papain solution in a vacuum orbital shaker under refrigeration at 4 °C for 24 h, the corresponding values were 1.3, 7.87, and 6.73 mg/100 g, respectively. All experimental treatments except for rehydration after hot-air drying (44.1 mg/100 g) resulted in values for TVBN below 25 mg/100 g; therefore, any of these treatments will keep squid mantle in acceptable fresh condition. The high TVBN value of hot-air-dried samples may be due simply to dehydration with concomitant concentration of the volatile nitrogen compounds. In an earlier study using jumbo squid [14], a similar increase was associated with decomposition of the salt ammonium chloride ($NH_4Cl$) during hot-air drying, which, in combination with moisture reduction, facilitated the concentration of TVBN by microbes; with a basal value of 14.67 mg/100 g, this effect was greater when drying was done at 90 °C (73.68 mg/100 g) than at 50 °C (29.38 mg/100 g).

### 3.4. Histological Observations by Light Microscopy

Histological microphotographs of control and variously-softened squid mantle (Figure 2) showed differences. Raw squid mantle exhibited circular muscles with fibers of regular shape, uniform size, narrow intervals, and thick diameter. The intracellular material was intact and the cells were in tight contact with each other (Figure 2a). After injection with 0.3% papain, the tissue showed severe fracturing, other damage, and non-uniform large spaces between now-loose muscle fibers (Figure 2b). This may have been due to the non-uniform, 1 cm-spaced injection pattern of the papain. In contrast, squid mantle soaked in 1.0% papain for 12 h while being subjected to vacuum orbital shaking evidently experienced enhanced enzyme infiltration into the tissues and hydrolysis of the muscle fibers; Figure 2c shows large, uniform spaces between the fibers, although here, too, the tissue showed severe damage in the form of fractures within loose muscle fibers. The softening effect of this treatment was obvious, and squid mantle so treated was the softest produced during this study (Table 4). Squid muscle soaked in 1.0% papain and subjected to ultrasonic processing for 30 min was only slightly harder ($p > 0.05$) (Table 4) and showed similar histological changes to the treatment above while creating smaller spaces between the cells (Figure 2d); the circular muscles were still visible. This latter process can shorten the necessary processing time to 30 min, versus 12 h for vacuum orbital shaking. Squid mantle subjected to $NaHCO_3$ treatment combined with ultrasonic cleaning showed uniform spaces, fractures, other damage, and loose muscle fibers (Figure 2e); circular muscles were not visible. These structural changes resulted in a softening of the texture of the squid mantle. The microstructure of muscle tissue rehydrated after hot-air drying showed severe shrinkage and compaction, with no hint of circular muscles (Figure 2f), and few spaces were found in this compacted tissue (Figure 2F). The cells were twisted, fibers showing different directions of striation crossed each other, and disordered, denatured myofibrillar protein was observed (Figure 2f). Vega-Galvez et al. [14] had already reported similar significantly disordered and denatured myofibrillar muscle fibers in meat of jumbo squid air-dried at temperatures of 50 and 90 °C.

### 3.5. SDS-PAGE

The protein components of raw and variously softened samples of squid mantle muscle were successfully analyzed by SDS-PAGE (Figure 3). Proteins extracted from raw squid muscle were mainly heavy-chain myosin (205 kDa). Hot-air drying caused this protein to degrade extensively, and 102 kDa and 57 kDa proteins followed the same pattern as myosin. The 172 kDa band, the prominence of which changed in a way opposite to that of myosin, probably consisted of degraded myosin. Increases of 90 kDa and 43 kDa proteins after treatment might represent fragments or subunits disassociated from other high-molecular-weight proteins. SDS-PAGE analysis had already shown that four kinds of muscle in the squid mantle were constituted of three major myofibrillar proteins: heavy-chain myosin of about 250 kDa, paramyosin of about 90 kDa, and actin of about 40 kDa [26]. Heavy-chain myosin (>180 kDa) was, however, easily degraded and was not found in either the control samples or samples subjected to various softening treatments (Figure 3), instead being represented by a new band at 130 kDa. Squid mantle soaked in 1.0% papain with vacuum orbital shaking (Figure 3, line (d)) showed obvious myosin degradation, with no protein integrity bands being evident at all on the SDS-PAGE gel. This result is consistent with our finding that squid mantle soaked in 1% papain with vacuum orbital shaking was the softest product of any experimental treatment (Table 4).

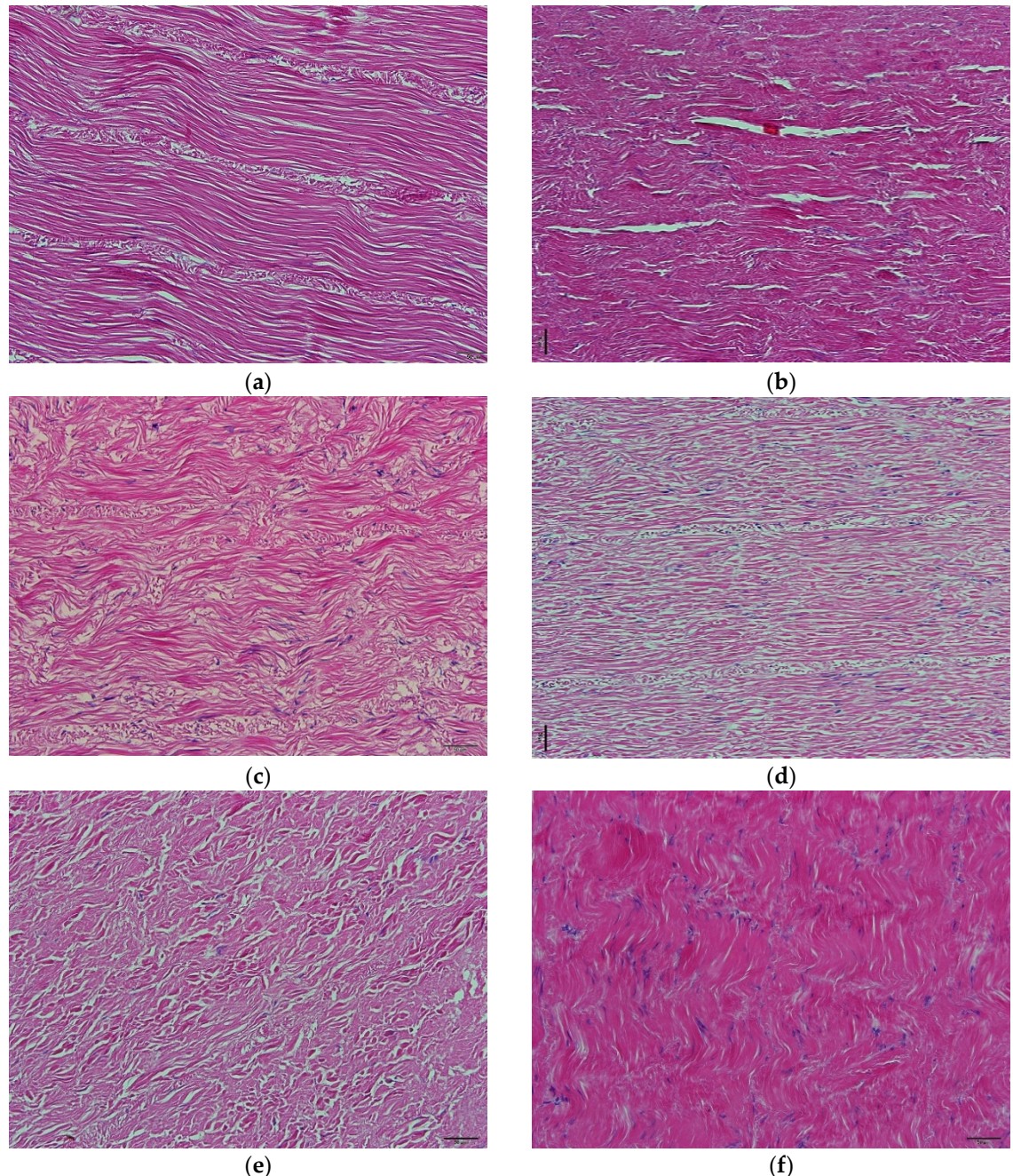

**Figure 2.** Histological sections of neritic squid mantle muscle (all × 200): (**a**) raw; (**b**) injected with papain; (**c**) soaked in papain with vacuum orbital shaking; (**d**) soaked in papain with ultrasonic processing; (**e**) soaked in NaHCO₃ solution with ultrasonic cleaning; (**f**) rehydrated after hot-air drying.

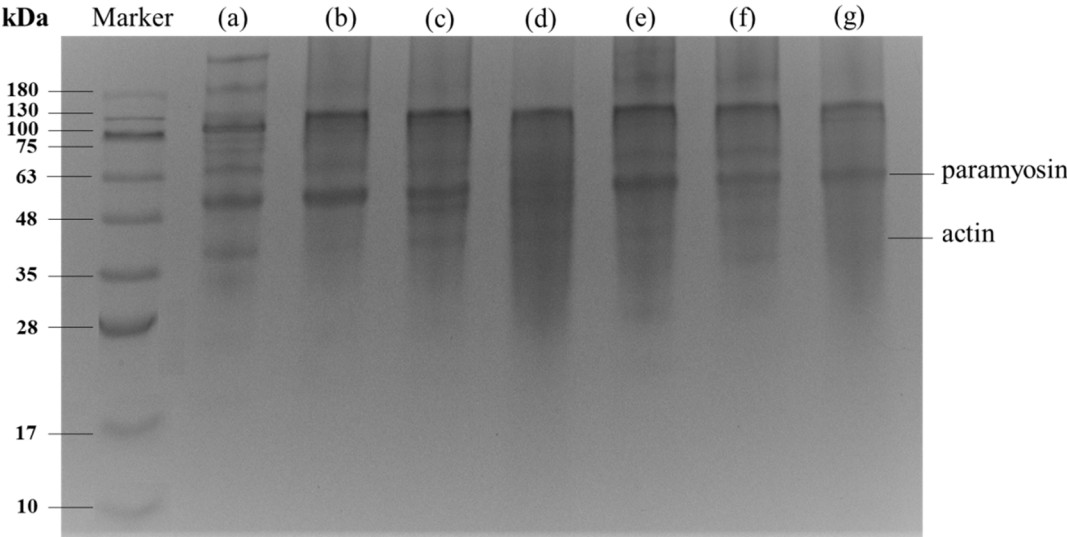

**Figure 3.** SDS-PAGE analysis of proteins in neritic squid fillets subjected to different softening methods: (**a**) raw squid; (**b**) control: untreated sterilized sample; (**c**) injected with 0.3% papain solution; (**d**) soaked in 1.0% papain solution in vacuum orbital shaker; (**e**) 1.0% papain solution in ultrasonic processor; (**f**) soaked in 2% NaHCO$_3$ solution with ultrasonic cleaning; (**g**) rehydrated after hot-air drying.

### 3.6. Texture Analysis

Hardness of squid muscle decreased significantly following sterilization (Table 4). Protein denaturation caused by the heat of sterilization decreased the proportion of contracted muscle fiber in the tissue and its water-holding capacity, resulting in a harder texture [13], but with no significant increase in hardness when dried at 60 °C as compared to 50 °C ($p > 0.05$). The texture-softening effect of hot-air drying and rehydration was not as effective as alkaline and enzyme treatments.

### 3.7. Sensory Evaluation

Squid mantle exposed to 2% alkaline solution developed a terrible fishy smell, although it was indeed tenderized by this treatment. Lest the sensory evaluation scores by human subjects be distorted, samples soaked in 2% NaHCO$_3$ and subjected to ultrasonic cleaning were used instead. Controls had significantly higher ratings (uniformly "extremely like") for color, texture, appearance, overall acceptability, and fishy smell ($p < 0.05$) than any treated squid (Table 8), whereas among the treated squid those soaked in 2% NaHCO$_3$ and subjected to ultrasonic cleaning were ranked second for these attributes (Table 8). Only with regard to flavor were the scores of controls and this particular treatment statistically different ($p > 0.05$; Table 8). Looking only at the scores for panelists over 70 years of age, there were no differences between the scores for texture and overall acceptability between controls and this particular treatment.

No panelists, not even those over 70 years of age, had any problem with chewing and eating control samples, but they felt that the texture of treated squid was mushy and its flesh was too pale. They preferred chewy squid over mushy squid and expected that seafood should have a texture resistant to chewing. It would be valuable to check whether elderly people with difficulties in eating and swallowing might prefer softened seafood. In our study, softness of squid muscles showed a negative relationship with the concentration of both injected enzyme and also with the concentration of soaked enzyme when soaking was done in either an ultrasonic processor or a vacuum orbital shaker (Tables 2–4). Whatever the experimental treatment, hardness also decreased with increasing enzyme concentration, so an overdose of the latter would be likely to downgrade the flavor of squid mantle (Tables 2–4). NaHCO$_3$ at 2% combined with ultrasonic cleaning and subsequent washing out of the

alkaline solution appears to be the most feasible process for softening mantles of neritic squid *Uroteuthis edulis* while retaining palatability.

**Table 8.** Sensory evaluation of neritic squid by panelists. Results for each feature are expressed as sums of individual scores ranging from 1 to 4.

| Treatments: | Control | Enzyme | | Alkali Soaking |
| --- | --- | --- | --- | --- |
| Features | | Injection | Vacuum Orbital Shaker | Ultrasonic Cleaning |
| All Panelists (*n* = 51), Mean Age 67 years | | | | |
| Color | 73 [a] | 174 [c] | 149 [c] | 114 [b] |
| Texture | 92 [a] | 149 [c] | 149 [c] | 120 [b] |
| Appearance | 72 [a] | 170 [c] | 164 [c] | 104 [b] |
| Flavor | 90 [a] | 152 [b] | 158 [b] | 111 [a] |
| Overall acceptability | 72 [a] | 162 [c] | 168 [c] | 105 [b] |
| Fishy smell | 94 [a] | 146 [b] | 147 [b] | 123 [b] |
| Older Panelists (*n* = 21), Age > 70 years | | | | |
| Color | 20 [a] | 59 [c] | 49 [c] | 42 [b] |
| Texture | 33 [a] | 46 [a] | 44 [a] | 47 [a] |
| Appearance | 22 [a] | 55 [b] | 51 [b] | 42 [b] |
| Flavor | 28 [a] | 52 [b] | 48 [b] | 42 [a] |
| Overall acceptability | 26 [a] | 56 [b] | 50 [b] | 38 [a] |
| Fishy smell | 31 [a] | 50 [b] | 45 [b] | 44 [b] |

Values followed by the different superscript letters within each row are significantly different ($p < 0.05$). Control: untreated; injection: injected with 0.3% papain solution; vacuum orbital shaker: soaked in 1.0% papain solution in vacuum orbital shaker; ultrasonic cleaning: soaked in 2% NaHCO3 solution in ultrasonic cleaner.

## 4. Conclusions

The results of this study showed that sterilized mantle slices of neritic squid *Uroteuthis edulis* soaked in 2% $NaHCO_3$ in an ultrasonic cleaner became significantly softer than control samples while retaining a similar texture and overall acceptability, as judged by elderly panelists. Soaking jumbo squid in $NaHCO_3$ solution and washing with tap water has been used by some kitchens to rehydrate and modify its texture. We recommend incorporating ultrasonic cleaning as well, in order to accelerate the softening effect on the squid. Histological observation showed that, depending on the process, softening occurred due to different kinds of degradation of the circular muscle fiber structure. Although squid mantle soaked in 1% papain in a vacuum orbital shaker produced the softest end product of all the treatments, which was confirmed both by histology and texture analysis, the panelists considered such squid to be unpalatably mushy. Nonetheless, it could provide an alternative for seniors who with masticatory impairment. Such information may benefit the food industry in terms of developing and processing food for senior citizens.

**Author Contributions:** Conceptualization, W.-C.S. and Y.-W.F.; writing—original draft preparation, M.J.G. and W.-C.S.; writing—review and editing, M.J.G., methodology, W.-C.S., and Y.-W.F.; formal analysis, Y.-W.F.; investigation, W.-C.S. and Y.-W.F.; resources, Y.-W.F. and W.-C.S. All authors have read and agreed to the published version of the manuscript.

**Funding:** This research was funded by the Fisheries Agency of the Republic of China (Taiwan) (108AS-3.4.1-FA-F1).

**Acknowledgments:** The authors are grateful for the funding provided by the Fisheries Agency of the Republic of China (Taiwan) (108AS-3.4.1-FA-F1). M.J.G.'s work as a Research Fellow at National Taiwan Ocean University's Center of Excellence for the Oceans depended on continued support for the center by the Featured Areas Research Center Program within the Taiwan ministry of Education's Higher Education Sprout Project.

**Conflicts of Interest:** The authors declare no conflict of interest.

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
