# Peer review of "Effects of Different Softening Processes on the Hardness and Quality of Thawed Neritic Squid (Uroteuthis edulis) Muscle"

_processes, doi:10.3390/pr8020135_

Round 1

Reviewer 1 Report

I have reviewed the manuscript titled; Effects on the Hardness and Quality of Neritic Squid (Loligo edulis) Muscle by different Softening Processes

Comments and Suggestions for Authors

Title should be rewritten:

-should be added- thawed

This article aims to evaluate the  better tenderizing methods than either enzyme treatment or hot-air drying for neritic squid mantle intended for consumption by senior citizens.

The work is relevant and topical but there are many  aspects of the manuscript that could be improved upon to ensure the manuscript is acceptable  to print.

Although the article is not innovative, it contains original and interesting information.

This article would be improved if the authors clarify or revise the following:

English should be revised by a native English speaker. Although no major mistakes were detected and the manuscript can be easily understood, minor mistakes should be revised.

Lines 38.- 39. „Squid mantle muscle fibers, arranged both radially  and circularly, are supported by connective tissue that is oriented in radial, circular, and longitudinal  directions“,- references needed. Reference 4 is not for sentence  „This has made it difficult to adapt squid mantle into a softer processed food for older people“.

Lines 69. - 76. - need rows spacing.

Lines 80. – 81. „………after harvest was  purchased from the Keelung Fishermen’s Association (Keelung, Taiwan).......“ -there should have been a diagram to illustrate these sampling regions as is done in such works. Date or season of harvest? 

Lines  132.-140. Total Volatile Basic Nitrogen (TVBN)

Comment:using some tests for chilled marine products to frozen marine products is not correct; the results will then indicate broadly the state of spoilage of the squid before freezing, while additional tests may be applied to estimate the degree of deterioration during frozen storage. 

Line 133. „Squid mantle strips were homogenized……… „ mantle with skin?

Lines 180.-182. „2.8 Sensory Evaluation“-  is not clear, need clarification. What method was used? Torry is for cooked flavour of squid. Sample size is very important for sensory evaluation because common mantle length in commercial catches ranges between 15 and 25 cm, the muscle structure depends on the size - NO DATA

Samples size are very important for complete  research.

Line 415. Ref 14. „Moreover, the squid size is an important factor that affects directly its ammonia content. Big sized  squid (adults) samples were related to high contents of TVBN (from 70 to 270 mg N/100 g sample), (Albretch, 2006). However, it was reported that the acceptable limit of squid TVBN was dependent on the species,environment and physiological conditions, processing and storage conditions and thus, highly variable (Fu et al., 2007; Quitral Robleset al., 2003)“

Line 2014. Remove the extra dot and font is too large.

Lines 216.-2017. „Why did treatment with NaOH or NaHCO3 result in a significant decrease in squid  hardness?“ – Please delete.

Line 250. Tbl font is too large.

Figure 2. font is too large.

Lines 260.- 265. -„Raw squid mantle had a moisture content of 80.9%, together with 14.4% protein, 1.1% fat, 3.4%,  ash, and 0.2% carbohydrate, and these proportions did not change appreciably after sterilization at  121°C for 20 min………………………….“ THERE ARE NO METHODS DESCRIPTIONS!

Line 274. ……..for chilled products  Line 80. „Neritic squid (Uroteuthis edulis) that had been frozen on board fishing boats………….“ This is problem, authors were used the wrong method in this experiment.

Line 415. Reference 14…………..“the squids were maintained refrigerated..“

Lines 365.- 376. The conclusions are very poor without facts from experiments. They should be strengthened. The sentences needs rewriting. The conclusions has no value in present form.

References needed for effects on the gardness and quality of thawed Cephalopods ( squids).

Author Response

Ref. No.: processes-688555

Original Title: Effects on the Hardness and Quality of Neritic Squid (Loligo edulis) Muscle by different Softening Processes

Author(s): Mark J. Grygier, Yu-Wen Fan and Wen-Chieh Sung

To: Referee 01

We have revised the paper based in part on the referee’s comments. The following table provides a comparison between the referee’s comments and the authors’ revision. Textual revisions are all shown in red in the revised manuscript.

Referee’s comments

Resulting revision

Title should be rewritten:

-should be added- thawed

We thoroughly revised the title, which now reads, “Effects of Different Softening Processes on the Hardness and Quality of Thawed Neritic Squid (Uroteuthis edulis) Muscle”.

1.      English should be revised by a native English speaker. Although no major mistakes were detected and the manuscript can be easily understood, minor mistakes should be revised.

The text, tables, and figure captions, including all added, revised, and reorganized parts, has been rechecked carefully by the first author, an American zoologist with vast copy-editing experience.

2.      Lines 38.- 39. “Squid mantle muscle fibers, arranged both radially and circularly, are supported by connective tissue that is oriented in radial, circular, and longitudinal directions”,- references needed. Reference 4 is not for sentence “This has made it difficult to adapt squid mantle into a softer processed food for older people”.

We moved the citation of reference [4] to follow the anatomical information, which is lightly paraphrased from that paper.

3.      Lines 69. - 76. - need rows spacing.

The spacing has been adjusted to match the rest of the text.

4.      Lines 80. – 81. “… after harvest was purchased from the Keelung Fishermen’s Association (Keelung, Taiwan) ...” -there should have been a diagram to illustrate these sampling regions as is done in such works. Date or season of harvest? 

We now mention that squid fishing took place from June to September in 2018, but since our paper does not concern the squid fishery per se, we do not think it necessary to add a map showing where Keelung and Taiwan are located in Asia or the Western Pacific. Also, since the precise sites of squid capture in the vicinity of Keelung are unknown, it is not possible to provide a map showing them.

5.      Lines 132.-140. Total Volatile Basic Nitrogen (TVBN)

Comment: using some tests for chilled marine products to frozen marine products is not correct; the results will then indicate broadly the state of spoilage of the squid before freezing, while additional tests may be applied to estimate the degree of deterioration during frozen storage. 

We did not use fresh squid in this study, partly because there might be various uncontrolled differences between fresh squid obtained on different dates, and between such squid and frozen squid, but mainly because all squid are frozen on board the fishing boats, and any unfrozen squid sold in the market has been thawed. As for estimating the degree of deterioration, please see our answer to point 8 below.

6.      Line 133. “Squid mantle strips were homogenized …” mantle with skin?

We have newly specified “without skin” at several places in the Materials and Methods.

7.      Lines 180.-182. “2.8 Sensory Evaluation”-  is not clear, need clarification. What method was used? Torry is for cooked flavour of squid. Sample size is very important for sensory evaluation because common mantle length in commercial catches ranges between 15 and 25 cm, the muscle structure depends on the size - NO DATA

Samples size are very important for complete research.

The mantle length of neritic squid is at least 12 cm. For each experimental treatment and control, two squid strips about 1.5 cm wide were offered with 2 dips of soy sauce to each panelist; this is now fully explained in section 2.8.

8.      Line 415. Ref 14. “Moreover, the squid size is an important factor that affects directly its ammonia content. Big sized squid (adults) samples were related to high contents of TVBN (from 70 to 270 mg N/100 g sample), (Albretch, 2006). However, it was reported that the acceptable limit of squid TVBN was dependent on the species, environment and physiological conditions, processing and storage conditions and thus, highly variable (Fu et al., 2007; Quitral Robleset al., 2003)”

The reviewer seems to imply that it is improper to assay TVBN in our samples because many factors can influence the outcome, and that we should cite the quoted works. We instead try to address this objection indirectly by pointing out that the mantle length of neritic squid harvested around Keelung in the summer of 2018 was consistently around 12-20 cm and that squid frozen on board fishing boats was sold within half a year. We never detected concentrations of TVBN exceeding 50 mg/100 g in any treated sample, so any deterioration of the squid between harvest and thawing was apparently minimal.

9.      Line 2014. Remove the extra dot and font is too large.

We deleted the extra period and reduced the font size to match that of other tables.

10.  Lines 216.-2017. “Why did treatment with NaOH or NaHCO3 result in a significant decrease in squid hardness?” – Please delete.

This rhetorical question has been deleted.

11.  Line 250. Tbl font is too large.

Figure 2. font is too large.

Font size here and elsewhere has been adjusted to be consistent throughout the manuscript.

12.  Lines 260.- 265. –“Raw squid mantle had a moisture content of 80.9%, together with 14.4% protein, 1.1% fat, 3.4%,  ash, and 0.2% carbohydrate, and these proportions did not change appreciably after sterilization at  121°C for 20 min…” THERE ARE NO METHODS DESCRIPTIONS!

At the end of section 2.2 we added a statement to the effect that AOAC procedures were followed. [We also added a description of the sterilization procedure to section 2.8.]

13.  Line 274. ……..for chilled products  Line 80. “Neritic squid (Uroteuthis edulis) that had been frozen on board fishing boats…” This is problem, authors were used the wrong method in this experiment.

Line 415. Reference 14…………..“the squids were maintained refrigerated…”

We added the sentence “Freshly caught, unfrozen squid were not available for analysis, but since only frozen or frozen-and-thawed squid are sold on the open market, our use of raw, newly thawed squid as controls was justified by a need for consistency.”

We also changed “refrigerated” to “under refrigeration”.

14.  Lines 365.- 376. The conclusions are very poor without facts from experiments. They should be strengthened. The sentences needs rewriting. The conclusions has no value in present form.

We rewrote the Conclusions section, with a partial change of emphasis, so as to provide direct advice to the food industry.

15.  References needed for effects on the hardness and quality of thawed Cephalopods (squids).

We added to the end of the Introduction a summary of the findings of Gokoglu et al. (2018) on this point (new Reference [15]), with respect to a different species of squid, Loligo vulgaris.

Reviewer 2 Report

I enjoyed reading this manuscript; the needs of special groups of seafood consumers include the elderly people. This manuscript presents some interesting data. The protocols include biochemical, histological and organoleptic evaluation of different methods of processing squids. The results of this work could be used by the seafood industry and would help the preparation of specially treated seafood products.

I have two suggestions to make and it is about the histological work.

1) It would be helpful to have the same orientation of all the histological sections presented in Fig 2.

2)The authors say that a “…number of 2-3 mm thick strips of squid mantle”  I wonder if they had similar size of all different treatments. I guess they can easily control for this if they have a few sections of a few different tissue strips from each treatment. Did they trim the tissue to a smaller size after embedding it and before histological sectioning?  These questions should be easy to answer and I expect that the authors can defend the protocol.

Author Response

Ref. No.: processes-688555

Original Title: Effects on the Hardness and Quality of Neritic Squid (Loligo edulis) Muscle by different Softening Processes

Author(s): Mark J. Grygier, Yu-Wen Fan and Wen-Chieh Sung

To: Referee 02

We have revised the paper based in part on the referee’s comments. The following table provides a comparison between the referee’s comments and the authors’ revision. Textual revisions are all shown in red in the revised manuscript.

Referee’s comments

Resulting revision

I have two suggestions to make and it is about the histological work.

1)      It would be helpful to have the same orientation of all the histological sections presented in Fig 2.

2)      The authors say that a “…number of 2-3 mm thick strips of squid mantle” I wonder if they had similar size of all different treatments. I guess they can easily control for this if they have a few sections of a few different tissue strips from each treatment. Did they trim the tissue to a smaller size after embedding it and before histological sectioning?  These questions should be easy to answer and I expect that the authors can defend the protocol.

1) We turned Figure 2 (b) and (d) 90 degrees all the histological sections have the same orientation.

2) The requested details of sectioning procedure have been added to section 2.5 [A number of 2-3 mm thick strips of squid mantle subjected to different softening techniques as outlined above were fixed with 10% formalin at 4°C for 24 hr. Dehydration and embedding were performed according to [8]. Samples were dehydrated in 70%, 90%, and 100% ethyl alcohol for 1.5 hrs each to remove free water in the fixed tissue, then subjected to 2 changes of xylene for 1 hr each, infiltration in melted paraffin at 60°C for 30 min, and then embedding in paraffin for 24 hrs. The block face was trimmed to 5 mm × 5 mm × 3 mm. Thick sections (6 μm) were prepared with a rotary microtome (Tissue-Tek VIP5 JR, Sakura, Tokyo, Japan) using a steel knife. Tissue sections mounted on glass slides were dried at 40°C [21] and immersed in 2 changes of xylene for 10 min each to dissolve the paraffin, then immersed in ethyl alcohol of decreasing concentration: 100%, 95%, 90%, 85%, 80%, and 75% for 5 min each. After washing with distilled and double distilled water, the slides were immersed in a saturated hematoxylin solution for 20 sec and washed under tap water for 15 sec. The slides were then stained by immersion in eosin (Thermo Fisher Scientific, Kalamazoo, MI, USA) for 45 sec, the excess being washed out in tap water. The slides were then dehydrated in ethyl alcohol of increasing concentration -- 75%, 80%, 85%, 90%, 95%, and 100%, for 10 sec each -- and then in xylene for 10 sec [21]. Then, glass cover slips were applied with mounting medium (Micromount, Leica Biosystems Richmond, Inc. Richmond, IL, USA). Observations were done with a compound photomicroscope (BX 53 Upright Microscope, Olympus, Melville, NY, USA) at 200x and 400x equipped with a digital camera (BX 53 Upright Microscope, Olympus, Melville, New York, USA).].

Round 2

Reviewer 1 Report

.................121°C for 20 min in an autoclave (TM-322, Tomin Medical Equipment Co., Ltd., New Taipei City, Taiwan).- see font size

.................Fisheries Agency of the Republic of China (Taiwan)  -font

Author Response

Dear Reviewer

The authors are extremely grateful to anonymous referee involved for providing his/her excellent comments and valuable advice in this paper, again. We have revised the paper based on the referee’s comments. We have pleasure in requesting the referee to review this paper. Thank you. Your prompt attention to this paper will be much appreciated.

Point 1:

.................121°C for 20 min in an autoclave (TM-322, Tomin Medical Equipment Co., Ltd., New Taipei City, Taiwan).- see font size

Response 1: 

We revised the font size of the above sentence of section 2.8 Sensory Evaluation at page 5. Sorry for the mistake and thanks for the comment. (Please see the revised manuscript).

Point 2:

.................Fisheries Agency of the Republic of China (Taiwan)  -font

Response 2:

The font size of Fisheries Agency of the Republic of China (Taiwan) has been revised. Thanks for pointing out the problem. (Please see the funding information of revised manuscript at page 15).

Thank you for your consideration.

Yours truly,

Wen-Chieh Sung, Ph.D.

Professor

Department of Food Science

National Taiwan Ocean University
